# Cognitive Predictors of Posttraumatic Stress in Children 6 Months after Paediatric Intensive Care Unit Admission

Belinda L. Dow [1],*[ID], Justin A. Kenardy [2], Robyne M. Le Brocque [3] and Debbie A. Long [1,4][ID]

1   Centre for Healthcare Transformation, School of Nursing, Queensland University of Technology, Brisbane, QLD 4059, Australia
2   School of Psychology, University of Queensland, Brisbane, QLD 4072, Australia
3   School of Nursing and Midwifery, University of Queensland, Brisbane, QLD 4072, Australia
4   Paediatric Intensive Care Unit, Queensland Children's Hospital, Brisbane, QLD 4101, Australia
*   Correspondence: belinda.dow@qut.edu.au

**Abstract:** This study aimed to identify predictors, especially cognitive predictors, of posttraumatic stress symptoms (PTSS) and posttraumatic stress disorder (PTSD) in children 6 months after Paediatric Intensive Care Unit (PICU) admission. Participants were 55 children aged 6–16, admitted to PICU for at least 8 h. Medical data were collected from patient charts. Cognitive variables (peri-trauma affect, cognitive processing and trauma memory) were assessed by interview and self-report questionnaires 2–4 weeks and 6 months following PICU admission. Acute PTSS at 2–4 weeks were assessed by self-report questionnaire and PTSD at 6 months was assessed by clinical interview. Receiving ketamine in PICU was the only non-cognitive variable associated with PTSS at 6 months. Peri-trauma affect, cognitive processing, and trauma memory significantly and independently accounted for 21% of the variance in PTSS at 6 months even after controlling for acute PTSS (and ketamine). A mediation analysis showed that peri-trauma affect indirectly influenced PTSS at 6 months through its effect on cognitive processing. Conclusions: Cognitive variables significantly contribute to PTSS in children, following PICU admission. Peri-trauma affect influenced PTSS only via disrupted cognitive processing. Prevention or early intervention strategies aimed at helping children develop a complete, contextual trauma narrative may be effective in reducing persistent posttraumatic stress responses in children following PICU.

**Keywords:** children; Intensive Care; PICU; posttraumatic stress disorder; risk factors; memory





## 1. Introduction

More than 13,000 children are admitted to Paediatric Intensive Care Units (PICUs) in Australia and New Zealand each year [1]. In addition to the physical repercussions of critical illness and related treatment, up to 28% of children report elevated posttraumatic stress symptoms (PTSS) in the acute aftermath of PICU discharge, and around 1 in 4 will meet posttraumatic stress disorder (PTSD) criteria by 6 months post-discharge [2]. Persistent posttraumatic stress responses have been associated with emotional dysfunction, adverse physical health outcomes and poorer health-related quality of life [3–5]. The identification of risk factors that trigger or maintain distress is important for screening and the development of prevention and targeted intervention strategies to support children and families after PICU discharge.

Few consistent pre-morbid and medically-related PTSD risk factors have been identified in the literature. Significant post-traumatic stress following PICU may be associated with premorbid psychopathology [6], non-elective admission status [7] and parental PTSS and psychopathology [6,8,9]. Age, gender and other demographic factors do not appear to be associated with PTSS, nor do admission variables such as diagnosis, duration of mechanical ventilation and duration of opiates/benzodiazepines [10]. Most studies have failed to find an association between PTSS and objective measures of disease severity. Exposure to

invasive treatment procedures and treatment intensity may be associated with acute, but not long-term distress [11]. Conceptual models of psychological distress following trauma have directed researchers to more subjective factors that improve our ability to identify at-risk children after PICU.

Cognitive models (e.g., Ehler's and Clark [12]) propose that PTSD arises, in part, as a result of high levels of fear, dissociative reactions and a maladaptive cognitive processing during a traumatic event. The nature of the ongoing trauma memory will depend on how well the trauma was initially encoded. Data-driven processing (confused, disorganized processing of sights, sounds and smells associated with the event) as opposed to conceptual processing (clear, organized processing of meaningful aspects) will result in highly sensory trauma memories that are poorly elaborated and difficult to recall intentionally. Maladaptive coping strategies such as avoidance and rumination will maintain PTSD symptoms. Leaders in the field have suggested that cognitive models of PTSD in adults also apply to children [13,14].

Evidence supporting the role of peri-trauma affect and memory processes, such as encoding and recall of traumatic events, is emerging in children and adolescents. Peri-trauma fear has been associated with PTSS in children [15,16]. In one study, confusion and overall recall of an event approached significance as a predictor of PTSD 6 weeks post-trauma, but not PTSS at 8 months post-trauma [12]. Associations have also been reported between memories of a sensory nature and acute stress [16,17], and between data-driven processing and acute stress symptoms or PTSS [17–19]. This emerging evidence highlights the potential importance of peri-trauma affect and encoding in child PTSD. These factors may be particularly salient for critically ill children, as their encoding and recall of events occurring in the PICU are often highly compromised by the nature of their admission and treatment (e.g., sedation and other medications, injury/illness-related factors such as traumatic brain injury, neurosurgery, sepsis and delirium). In addition, Ehler's and Clark's [12] model suggests that peri-trauma affect could influence the development of PTSD directly, or indirectly through influencing cognitive processing during the event. Identifying whether peri-trauma affect, such as acute fear and panic, are associated directly with a risk of ongoing distress, or are perhaps partially or fully mediated by their influence on cognitive processing, is important for early screening, assessment and development of early intervention strategies.

In a previous paper, we reported that cognitive/affective factors including peri-trauma affect, cognitive processing and acute sensory memory quality were significantly associated with acute PTSS in school-aged children and adolescents 2–4 weeks following PICU discharge [20]. These variables significantly added to the prediction of acute PTSS over and above significant premorbid and trauma characteristics (age, admission for traumatic injury), increasing the explained variance from 18% to 38%. The purpose of the current study was to investigate whether acute cognitive factors are important in children's long-term distress following PICU admission. Specifically, this study followed up the sample described in Dow et al. [20] at 6 months post-discharge to (1) identify the predictors of PTSS in children 6 months post-PICU; (2) explore whether cognitive factors add to the prediction of PTSS at 6 months, over and above premorbid and trauma characteristics and (3) explore whether any association between peri-trauma affect and PTSS at 6 months is mediated by disrupted cognitive processing.

## 2. Materials and Methods

### 2.1. Sample and Setting

Participants were recruited as part of a wider prospective longitudinal study investigating the psychological impact of PICU admission on children. Families of surviving children aged 6–16, admitted to the Royal Children's Hospital PICU, Brisbane, Australia for at least 8 h from June 2008 to January 2011, were recruited consecutively. Exclusion criteria were: (a) any prior PICU admission; (b) admission > 28 days; (c) post-traumatic amnesia > 28 days; (d) non-accidental injury; and (e) developmental delay or intellectual impairment.

*2.2. Measures*

2.2.1. Premorbid and Trauma Characteristics

Demographic information and premorbid information were obtained via parent questionnaire. Medical data were obtained from patient charts. Data obtained included age at PICU admission, gender, prior trauma exposure (number of prior traumas), parent's report of child exposure to any distressing events in PICU (yes/no), whether another patient died in the unit during the child's admission (as a specific, potentially distressing event exposure for the child and/or family; yes/no), premorbid behavioural/emotional problems (yes/no), length of stay in PICU > 48 h (yes/no), admission for traumatic injury, number of invasive procedures, ventilation (yes/no), intubation (yes/no), received specific medication agent (midazolam, morphine, propofol, ketamine; yes/no). Illness severity was measured with the Revised Paediatric Index of Mortality (PIM2; Slater et al., 2003). The PIM2 is a regression model that uses admission data (e.g., blood pressure, diagnosis, mechanical ventilation) to predict intensive care outcome for children. Scores indicated risk of death at admission (%). For further detail of how trauma characteristics were measured and operationalised, see [20].

2.2.2. PTSS and PTSD

Acute PTSS (at 2–4 weeks post-PICU) were assessed with the self-report Children's Revised Impact of Event Scale (CRIES-13; 21). The scale provides a continuous score of PTSS, although a score of 30 or above represents a clinically elevated score. The CRIES-13 has acceptable internal consistency ($\alpha$ = 0.80–0.87) and convergent validity with other scales [21,22]. PTSD symptom severity and PTSD diagnosis were assessed using the Children's PTSD Inventory (CPTSDI; 23). The CPTSDI is a DSM-IV-based clinician-administered diagnostic interview assessing PTSD in youth aged 7–18 years. The CPTSDI is supported by strong psychometric properties, with excellent content validity and concurrent validity with other well-established PTSD interviews. Overall diagnostic agreement has been reported as 0.93–0.95 [23,24]. In this study, the total number of Criterion B, C and D symptoms endorsed on the CPTSDI were summed to create the main dependent variable "PTSS at 6 months." PTSD diagnosis was also calculated in accordance with an alternative algorithm (PTSD-AA; [25]) that has been shown to better identify children suffering distress and functional impairment [26].

2.2.3. Cognitive Variables

Several cognitive/affective items (where indicated) were derived from the Intensive Care Unit Memory Tool (ICUMT; [27]), completed during the acute assessment. The ICUMT assesses patients' recall of their admission. It was developed for use in adults, but it has been successfully used in a sample of PICU survivors aged 7–17 years [7]. The wording of some items was simplified or modified for use with children, as described in Dow et al., [20].

Peri-Trauma Affect

All items were from the ICUMT and were dichotomously scored. Peri-trauma fear was assessed by one item, "When you were in Intensive Care, did you feel scared, frightened or worried?" Peri-trauma panic was assessed by one item "When you were in Intensive Care, did you feel like you were going to panic?" Peri-trauma sadness was assessed by one item, "When you were in Intensive Care, did you feel sad or unhappy?"

Peri-Trauma Cognitive Processing

Items were from the ICUMT. Data-driven processing was assessed by one item (henceforth called Confusion), "When you were in Intensive Care, were you confused?" This item was used to assess data-driven processing by Ehlers et al., (2003) although their item was scored on a three-point scale. Peri-trauma delusions. Children were scored as experiencing

delusions if they endorsed any 'delusional' item from the ICUM (nightmares, weird dreams, hallucinations, paranoid delusions).

Acute Trauma Memory

Total PICU recall was assessed by one item, "How much do you remember of your stay in Intensive Care?" (rated on a 5-point scale from 'nothing' to 'everything'). Acute sensory memory quality was assessed by the Trauma Memory Quality Questionnaire (TMQQ; [28]). The TMQQ is an 11-item self-report measure that assesses the sensory quality and temporal context of children's trauma memories, and the extent to which the memories are verbally accessible. The scale has been used with children aged 7 years and older [17]. Higher scores indicate memories of a more sensory, less verbally accessible nature.

### 2.3. Procedure

Eligible families were invited to participate in the study upon discharge from PICU (*n* = 19; 35%), or by letter immediately following discharge (*n* = 36; 65%). Once written informed consent was obtained from parents and children aged 10 and older, families completed an acute assessment 2–4 weeks following PICU discharge (*Mdn* = 23 days, *R* = 9–46 days) and a follow-up assessment 6 months following PICU discharge (*Mdn* = 6.9 months, *R* = 5.4–8.7 months). At the acute assessment, parents and children completed questionnaires and children completed an interview (ICUMT). Interviews were conducted by phone or during outpatient visits. At 6-month follow-up, children completed questionnaires and a PTSD assessment via interview (CPTSDI). Interviews were conducted at the family home, by phone or during outpatient visits. This study was approved and conducted in accordance with the University of Queensland Medical Research Ethics Committee and the Royal Children's Hospital Human Research and Ethics Committee.

### 2.4. Data Analysis

All analyses were performed using the Statistical Package for Social Sciences (SPSS 19.0; Chicago, IL, USA). Bivariate correlations were conducted to identify associations between PTSS at 6 months and premorbid variables, trauma characteristics, and cognitive/affective variables. Bivariate correlations were also conducted between significant independent variables. Phi coefficients are reported for correlations between dichotomous variables; point biserial correlations are reported between dichotomous and continuous variables. Cognitive variables with high inter-correlations were combined to reduce the risk of over-inflating the regression model and to retain power.

Significant predictors were then entered into a hierarchical multiple regression analysis. Acute PTSS was entered in the first block, trauma characteristics in the second block and cognitive/affective variables were entered in the third block. The analysis was repeated in a logistic regression model with PTSD-AA diagnosis as the dependent variable.

Finally, a simple mediation analysis, conducted using ordinary least squares path analysis (via Hayes Process Macro for SPSS), was conducted to determine whether cognitive processing mediated the relationship between peri-trauma affect and PTSS at 6 months. We utilized a bootstrapping procedure [29] in which 95% CIs were used and 10,000 bootstrapping resamples were run.

## 3. Results

### 3.1. Demographics

Of 196 eligible families invited to participate, 70 children provided complete questionnaire and interview data 2–4 weeks following PICU discharge and of these, 55 children completed questionnaires and interviews 6 months following PICU discharge (15 dropouts: 4—unable to contact; 4—no concerns; 3—too busy; 2—child died; 2—incomplete data at 6 months). There were no significant differences between the final 55 participants and those who dropped out with regard to age (*t* = 1.71, *p* = 0.09), gender ($\chi^2$ = 1.15, *p* = 0.56), duration

of PICU admission ($t = -1.71$, $p = 0.09$), risk of death ($t = -1.28$, $p = 0.21$) or acute PTSS ($t = -1.99$, $p = 0.05$). Sample characteristics are presented in Table 1.

**Table 1.** Sample characteristics ($n = 55$).

| Characteristic | *n* (%) | *M* (SD) |
|---|---|---|
| Age, years | | 10.78 (2.65) |
| Gender, male | 32 (58%) | |
| Family of origin, both biological parents [a] | 38 (69%) | |
| Participating parent's highest level of education [a] | | |
|     Did not complete high school | 6 (11%) | |
|     Completed high school | 7 (13%) | |
|     College certificate | 16 (29%) | |
|     University degree | 19 (35%) | |
| Prior trauma exposure [a] (# of traumas) | | 1.39 (1.42) |
| Premorbid behavioural problems [a] | 5 (11%) | |
| Length of stay in PICU > 48 h | 12 (22%) | |
| PIM2 Risk of Mortality | | 1.81 (2.55) |
| Mechanically ventilated | 16 (29%) | |
| Reason for admission | | |
|     Post-operative care | 22 (40%) | |
|     Traumatic Injury | 13 (24%) | |
|     Respiratory | 6 (11%) | |
|     Other | 14 (25%) | |
| Admission status, elective | 21 (38%) | |
| Number of invasive procedures | | 4.98 (4.98) |
| Received therapeutic agents | | |
|     Midazolam | 13 (24%) | |
|     Morphine | 28 (51%) | |
|     Propofol | 9 (16%) | |
|     Ketamine | 13 (24%) | |
| Other patient death during admission | 5 (9%) | |
| Exposed to distressing event in PICU [a] | 3 (6%) | |
| Acute PTSS (>30 = elevated) | | 19.87 (18.00) |
| Peri-trauma affect | | |
|     Peri-trauma fear | 21 (38%) | |
|     Peri-trauma panic | 19 (35%) | |
|     Peri-trauma sadness | 21 (38%) | |
| Cognitive variables | | |
|     Confusion | 27 (49%) | |
|     Delusional experiences | 28 (51%) | |
|     Total PICU recall | | 1.91 (1.09) |
|     Sensory memory quality | | 23.75 (5.58) |

[a] Data missing for 6 families. # = Number.

### 3.2. Predictors of PTSS at 6 Months

The mean PTSS at 6 months score was 4.87 symptoms ($SD = 4.00$). Associations between PTSS at 6 months, and premorbid and trauma characteristics and cognitive variables are presented in Table 2. Receiving ketamine was the only significant trauma characteristic associated with PTSS at 6 months post-discharge. All of the peri-trauma affect variables were significant predictors, as were the cognitive processing variables and sensory memory quality.

Bivariate correlations between predictors are reported in Table 3. Significant associations were found between the three peri-trauma affect variables (fear, panic, sadness), and the two indictors of cognitive processing (confusion, delusions). These items were combined to form two factors for the following analyses.

Significant predictors were entered into a hierarchical multiple regression analysis. The model is presented in Table 4. To investigate whether significant predictors explained variance in PTSS at 6 months over and above their contribution to acute PTSS, acute PTSS was entered in the first step. It accounted for 22% of the variance in PTSS at 6 months

(adjusted $R^2$ = 0.21, $F$ (1, 53) = 14.78, $p < 0.001$). With the addition of ketamine at Step 2, the model accounted for 32% of the variance in PTSS at 6 months (adjusted $R^2$ = 0.30, $F$ (2, 52) = 12.00, $p < 0.001$). The cognitive variables were entered in the third step. With peri-trauma affect, cognitive processing and acute sensory memory quality at 6 months included at Step 3, the model accounted for 55% of the variance (adjusted $R^2$ = 0.50, $F$ (5, 49) = 7.75, $p < 0.001$). Each block significantly improved the prediction of PTSS at 6 months. Inspection of the final model indicated that the combined peri-trauma affect variable did not significantly contribute to the variance explained. Peri-trauma affect was highly correlated with disrupted cognitive processing (Spearman's ρ = 0.52, $p < 0.001$).

**Table 2.** Bivariate correlations between premorbid/peri-trauma/cognitive variables and (1) PTSS at 6 months and (2) PTSD-AA Diagnosis ($n$ = 55).

| Variables | PTSS at 6 Months | PTDS-AA Positive |
|---|---|---|
| **Premorbid factors** | | |
| Age | 0.02 | −0.05 |
| Gender | 0.08 | 0.11 |
| Prior trauma exposure [a] | 0.15 | −0.04 |
| Premorbid behavioural problems [a] | 0.23 | 0.09 |
| **Trauma characteristics** | | |
| *Disease-related* | | |
| Length of stay in PICU > 48 h | 0.02 | 0.05 |
| PIM2 Risk of death | 0.05 | 0.06 |
| Admission for traumatic injury | 0.01 | 0.02 |
| *Treatment-related* | | |
| Number of invasive procedures | 0.12 | 0.16 |
| Mechanically ventilated | 0.12 | 0.03 |
| Intubated | 0.08 | 0.01 |
| *Therapeutic agents* | | |
| Midazolam | 0.12 | 0.12 |
| Morphine | 0.22 | 0.23 |
| Propofol | 0.06 | 0.04 |
| Ketamine | 0.33 * | 0.21 |
| *Environment-related* | | |
| Other patient death during admission | 0.01 | 0.08 |
| Exposed to distressing event in PICU [a] | 0.03 | 0.05 |
| **Peri-trauma affect** | | |
| Peri-trauma fear | 0.28 * | 0.24 |
| Peri-trauma panic | 0.47 *** | 0.38 ** |
| Peri-trauma sadness | 0.27 * | 0.24 |
| **Cognitive variables** | | |
| *Peri-trauma cognitive processing* | | |
| Confusion | 0.45 ** | 0.33 * |
| Delusional experiences | 0.47 *** | 0.38 ** |
| *Acute trauma memory* | | |
| Total PICU recall | −0.13 | 0.02 |
| Sensory memory quality | 0.42 ** | 0.38 ** |

* $p < 0.05$, ** $p < 0.01$, *** $p < 0.001$. [a] data are missing for 6 families. Note: Point biserial correlations are reported between continuous and dichotomous variables. Phi coefficients are reported for correlations between PTSD-AA diagnosis and dichotomous variables.

**Table 3.** Bivariate correlations between significant independent variables ($n$ = 55).

| Variables | Acute PTSS | Peri-Trauma Fear | Peri-Trauma Panic | Peri-Trauma Sadness | Confusion | Peri-Trauma Delusions | Acute Sensory Memory Quality |
|---|---|---|---|---|---|---|---|
| Ketamine | −0.062 | −0.085 | 0.046 | −0.173 | 0.139 | 0.094 | −0.052 |
| Acute PTSS | - | 0.195 | 0.368 ** | 0.213 | 0.356 ** | 0.299 * | 0.437 *** |
| Peri-trauma fear | - | - | 0.452 *** | 0.461 *** | 0.501 *** | 0.260 | 0.077 |
| Peri-trauma panic | - | - | - | 0.531 *** | 0.204 | 0.412 ** | 0.359 ** |
| Peri-trauma sadness | - | - | - | - | 0.201 | 0.485 *** | 0.314 * |
| Confusion | - | - | - | - | - | 0.345 * | 0.157 |
| Peri-trauma delusions | - | - | - | - | - | - | 0.240 |

* $p < 0.05$, ** $p < 0.01$, *** $p < 0.001$. Note: Phi coefficients are reported for correlations between two dichotomous variables; Point biserial correlations are reported between dichotomous and continuous variables.

**Table 4.** Hierarchical regression analysis of PTSS at 6 months (CPTSDI total symptoms).

|  | B | SE B | β | $R^2$ | $\Delta R^2$ |
|---|---|---|---|---|---|
| Step 1 |  |  |  | 0.21 *** | 0.21 *** |
| Acute PTSS | 0.13 | 0.04 | 0.46 *** |  |  |
| Step 2 |  |  |  | 0.34 *** | 0.13 ** |
| Acute PTSS | 0.14 | 0.03 | 0.48 *** |  |  |
| Ketamine | 3.04 | 0.95 | 0.36 ** |  |  |
| Step 3 |  |  |  | 0.55 *** | 0.21 *** |
| Acute PTSS | 0.07 | 0.03 | 0.23 * |  |  |
| Ketamine | 2.81 | 0.83 | 0.33 ** |  |  |
| Peri-trauma affect | 0.41 | 0.36 | 0.14 |  |  |
| Peri-trauma cognitive processing | 1.39 | 0.49 | 0.33 ** |  |  |
| Acute sensory memory quality | 0.16 | 0.07 | 0.24 * |  |  |

* $p < 0.05$, ** $p < 0.01$, *** $p < 0.001$.

### 3.3. Predictors of PTSD-AA

In this sample, 16 (29%) patients met PTSD-AA criteria 6 months post-discharge. Associations between PTSD-AA, and premorbid and trauma characteristics and cognitive variables are also presented in Table 2. The pattern of significant and non-significant predictors of PTSD-AA was generally consistent with that of PTSS at 6 months, so the same model of independent predictors was used for PTSD-AA.

To determine the utility of the model in predicting PTSD diagnosis, a logistic regression analysis was conducted with PTSD-AA as the dependent variable. Results are presented in Table 5. At Step 1, acute PTSS significantly predicted PTSD-AA, (71% children correctly classified; $\chi^2 = 5.30$, $p = 0.02$). At Step 2, inclusion of ketamine did not significantly improve the model (78% of children correctly classified, $\chi^2 = 3.22$, $p = 0.07$). At Step 3, addition of the cognitive variables significantly improved the model and resulted in 80% of children correctly classified ($\chi^2 = 10.31$, $p = 0.02$). Inclusion of the cognitive variables in Step 3 resulted in a significantly better predictive model (Cox and Snell $R^2$ increased from 0.14 at Step 2 to 0.29 at Step 3; Nagelkerke $R^2$ increased from 0.21 to 0.41; sensitivity increased from 0.31 to 0.56).

**Table 5.** Hierarchical regression analysis of PTSD-AA.

| Independent Variables | B | SE B | OR | 95% CI for OR | $\chi^2$ STEP | $\chi^2$ MODEL | Correctly Classified | Cox & Snell $R^2$ | Nagelkerke $R^2$ |
|---|---|---|---|---|---|---|---|---|---|
| Step 1 |  |  |  |  | 5.30 * | 5.30 * | 71% | 0.10 | 0.10 |
| Acute PTSS | 0.06 | 0.03 | 1.06 * | 1.01–1.11 |  |  |  |  |  |
| Step 2 |  |  |  |  | 3.22 (*) | 8.53 * | 78% | 0.14 | 0.14 |
| Acute PTSS | 0.06 | 0.03 | 1.07 * | 1.01–1.13 |  |  |  |  |  |
| Ketamine | 1.30 | 0.73 | 3.65 (*) | 0.88–15.23 |  |  |  |  |  |
| Step 3 |  |  |  |  | 10.31 * | 18.84 ** | 80% | 0.29 | 0.29 |
| Acute PTSS | 0.02 | 0.04 | 1.02 | 0.95–1.10 |  |  |  |  |  |
| Ketamine | 1.32 | 0.86 | 3.74 | 0.69–20.33 |  |  |  |  |  |
| Peri-trauma affect | 0.26 | 0.35 | 1.30 | 0.65–2.58 |  |  |  |  |  |
| Peri-trauma cognitive processing | 0.83 | 0.51 | 2.30 | 0.84–6.29 |  |  |  |  |  |
| Acute sensory memory quality | 0.13 | 0.08 | 1.14 | 0.97–1.32 |  |  |  |  |  |

(*) $p < 0.10$, * $p < 0.05$, ** $p < 0.01$, OR = odds ratio.

### 3.4. Mediation Analysis

A mediation analysis was conducted to determine whether cognitive processing mediated the relationship between peri-trauma affect and PTSS at 6 months [29]. The analysis showed that peri-trauma affect indirectly influenced PTSS at 6 months, through its effect on cognitive processing. As shown in Figure 1 and Table 6, children who reported greater peri-trauma affect (fear, sadness and panic) also reported more disruptions to cognitive processing (confusion, delusions; $a = 0.3656$), and children reporting more disrupted cognitive processing reported greater PTSS at 6 months ($b = 1.8960$). A 95% bootstrap confidence interval for the indirect effect ($ab = 0.6931$) based on 10,000 bootstrap samples was entirely

above zero (0.2642 to 1.3485). There was no evidence that peri-trauma affect influenced PTSS at 6 months, independent of its effect on cognitive processing ($c' = 0.5911$, $p = 0.152$).

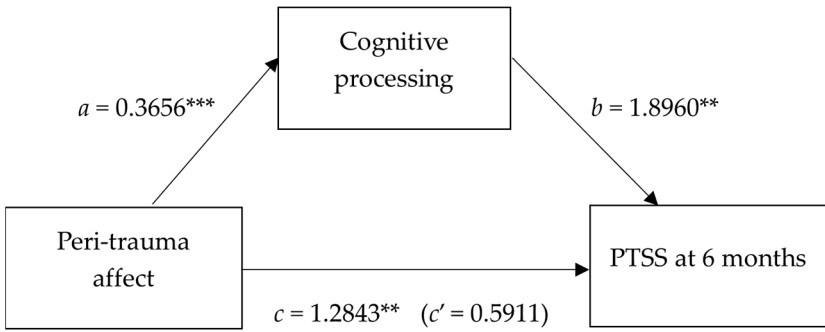

**Figure 1.** Simple mediation model for the relationship between peri-trauma affect and PTSS as mediated by cognitive processing. ** $p < 0.01$; *** $p < 0.001$.

**Table 6.** Simple mediation model coefficients for the relationship between peri-trauma affect and PTSS as mediated by cognitive processing.

| Antecedent | | \multicolumn{3}{c}{**Consequent**} | | | | |
|---|---|---|---|---|---|---|---|---|
| | | \multicolumn{3}{c}{**Cognitive Processing**} | | \multicolumn{3}{c}{**PTSS at 6 Months**} | |
| | | Coeff. | SE | $p$ | | Coeff. | SE | $p$ |
| Peri-trauma affect | $a$ | 0.3656 | 0.0844 | 0.0001 | $c'$ | 0.5911 | 0.4070 | 0.152 |
| Cognitive processing | | - | - | - | $b$ | 1.8960 | 0.5690 | 0.002 |
| Constant | $i_M$ | 0.5036 | 0.1362 | <0.001 | $i_Y$ | 2.4935 | 0.6327 | <0.001 |
| | | \multicolumn{3}{c}{$R^2 = 0.2613$} | | \multicolumn{3}{c}{$R^2 = 0.3210$} | |
| | | \multicolumn{3}{c}{$F(1, 53) = 18.74, p < 0.001$} | | \multicolumn{3}{c}{$F(2, 52) = 12.29, p < 0.001$} | |

## 4. Discussion

The purpose of this study was to identify predictors of PTSS and PTSD-AA at 6 months following PICU admission. In particular, this study sought to explore the role of cognitive factors in children's long-term posttraumatic stress. This study makes a valuable contribution to the literature, as there are few existing longitudinal studies exploring the factors associated with the onset and maintenance of PTSS in children following PICU. Few premorbid or trauma characteristics predicted children's PTSS or PTSD-AA at 6 months post-PICU. Receiving ketamine during admission was the only significant trauma characteristic. Ketamine is a dissociative anesthetic, usually administered intravenously with morphine, often being the first-line, preferred analgesic. Is it often used PRN for specific procedures (e.g., burns dressings), or as a continuous infusion for secondary management of pain. Reported emergence reactions include distortions in perceptions of sight and sound, alterations in mood, feelings of detachment from the environment and self, hallucinations, delirium and amnesia [30]. Age and admission for traumatic injury, which were found to be associated with children's acute PTSS in a previous paper [20], were not associated with children's PTSS at 6 months. This is consistent with findings of Le Brocque and colleagues [31] that younger children are more likely to show elevated PTSS that remit over time, compared to showing a stable, resilient trajectory. Objective measures of disease severity and other trauma characteristics were not predictive of PTSS at 6 months, consistent with findings in other paediatric patients [32].

In contrast, several cognitive variables were associated with PTSS and PTSD-AA diagnosis. Together, peri-trauma affect, peri-trauma processing and acute sensory memory quality significantly predicted PTSS and PTSD-AA 6 months post-trauma, even after controlling for PTSS 2–4 weeks post-trauma. Acute PTSS and ketamine accounted for 34% of the variance in PTSS at 6 months. Addition of the cognitive variables increased the predictive power of the model to 55% accountable variance. These results were mirrored for

the prediction of PTSD-AA diagnosis. Acute PTSS and ketamine resulted in 78% of children correctly classified. Addition of the cognitive variables slightly increased the predictive value of the model and resulted in 80% of children being correctly classified. The sensitivity of the model increased from 0.31 to 0.56, suggesting the final model was better at detecting distress where it was present.

Consistent with Ehlers and Clark's model [12], these results suggest that the quality of cognitive processing during a traumatic experience plays a significant role in child PTSS up to at least 6 months post-trauma. Peri-trauma processing predicted PTSS at 6 months, after controlling for acute PTSS and trauma memory. Peri-trauma affect was also predictive of PTSS at 6 months, although the relationship was fully mediated by peri-trauma cognitive processing. This suggests that fear, panic and sadness experienced during an event influences PTSS only by influencing children's encoding or memory of the event. This is consistent with evidence in adults that arousal during encoding appears to narrow attention such that only select cues are attended to and there is a failure to integrate contextual cues [33]. Sensory memory quality in the acute phase post-trauma was also found to be involved in the onset and maintenance of PTSS over time. Thus, children who leave PICU with sensory and fragmented memories (and those who received ketamine) are at greater risk of persistent or long-term PTSS/PTSD.

The results of this study, together with our previous findings relating to children's acute distress following PICU admission, highlight the importance of cognitive factors in the initial development and maintenance of posttraumatic stress in children following PICU admission. Given the high prevalence of PTSS and PTSD-AA in children following PICU admission, the identification of such factors indicates the need for prevention and intervention strategies to help school-aged children and adolescents cognitively accommodate their experiences in PICU and the consequences of their admission. Early strategies in PICU, or shortly after discharge, might aim to promote better peri-trauma processing and/or promote contextual understanding and integration of trauma memories. For example, where possible, providing information to children prior to or during PICU admission about their illness and treatment may encourage more meaningful peri-trauma processing, although there may be limited opportunity for this when children are sedated or highly medicated. Opportunities for at-risk children to discuss their admission shortly after discharge, and gain more contextual information about their PICU admission, may promote more adaptive, meaningful trauma memories. This may be supported by the use of a personal storybook about their PICU admission, specific advice for parents on how to talk to their children about the admission, or processing through drawing and play for younger children (with parents correcting misunderstandings, delusions, hallucinations). As trauma memory may be involved in maintaining PTSS over time, children who experience ongoing posttraumatic stress may benefit from exposure therapy with a focus on developing complete, factual trauma narratives that are less sensory and more verbally accessible in nature.

This study has a number of limitations that should be acknowledged. Firstly, the sample size is modest and replication of the current findings in larger samples is required. Furthermore, the participation rate suggests that these findings may not be representative of the entire PICU population. Although no significant differences were found between those who completed the acute assessment and those who dropped out, there was a trend for children with lower levels of acute distress to withdraw from the study prior to follow-up. Secondly, data-driven processing (confusion), peri-trauma fear, panic and sadness were assessed by single, dichotomous items. As noted by Ehlers et al. [12], the correlations between single items and PTSS are likely to underestimate the true relationship, due to increased error variance from measurement error. Thirdly, our primary focus was to explore encoding and trauma memory and we did not assess children's appraisals, which are a key component of Ehlers and Clark's [20] model. We also did not report on parental PTSS as a potential risk or maintaining factor, instead focusing on child factors. These represent opportunities for further research. Finally, it should be acknowledged that the importance

of cognitive factors, as found in this study, may not be applicable to younger children, given the likely variations in the cognitive, language and memory abilities of children aged under 6 years.

## 5. Conclusions

In summary, the results of this follow-up study of PICU survivors show that receiving ketamine in PICU, peri-trauma affect, peri-trauma cognitive processing and acute trauma memory quality significantly contribute to children's PTSS at 6 months, even after accounting for the influence of acute PTSS. These results suggest that preventative or early intervention strategies, aimed at helping children develop a complete, contextual trauma narrative that is less sensory in nature, more easily verbalized and more amenable to change once further information is available, may be effective in reducing chronic posttraumatic stress responses in children following PICU.

**Author Contributions:** Conceptualization, B.L.D., J.A.K., R.M.L.B. and D.A.L.; methodology, B.L.D. and J.A.K. writing—original draft preparation, B.L.D.; writing—review and editing, B.L.D., J.A.K., R.M.L.B. and D.A.L.; supervision, J.A.K., R.M.L.B. and D.A.L.; project administration, B.L.D. and R.M.L.B. All authors have read and agreed to the published version of the manuscript.

**Funding:** This research was funded by a Royal Children's Hospital Foundation "Near Miss" Research Grant (Grant no. 10261). B.L.D. was also supported by an Australian Postgraduate Award, Royal Children's Hospital Foundation Top-Up Scholarship, CONROD Top-Up Scholarship and the UQ Joseph Sleight Bursary.

**Institutional Review Board Statement:** The study was conducted in accordance with the Declaration of Helsinki, and approved by the University of Queensland Human Research Ethics Committee ([HREC], Ref: 20080000979), and the Royal Children's Hospital research ethics committee (20080000979).

**Informed Consent Statement:** Informed consent was obtained from all subjects involved in the study.

**Data Availability Statement:** The data presented in this study are available on request from the corresponding author. The data are not publicly available consistent with the Information Privacy Act (Qld) 2009, the Privacy Act (Cth) 1988, and the ethical standards outlined in the National Statement on Ethical Conduct in Human Research (2007)—Updated 2015.

**Acknowledgments:** The authors would like to thank all families who participated in this research study.

**Conflicts of Interest:** The authors declare no conflict of interest.

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
