# Peer review of "Cognitive Predictors of Posttraumatic Stress in Children 6 Months after Paediatric Intensive Care Unit Admission"

_traumacare, doi:10.3390/traumacare3020009_

Round 1

Reviewer 1 Report

I thank the authors for the opportunity to read this interesting paper.

Allow me to make a series of comments to the authors, especially referring to the materials and methods section.

This section should describe how the sample size was established and the sample was selected, but the results of the sample should be included in the results section. Thus, the data that appears between the lines of the text 101 to 108 and Table 1 must be detailed in the results section.

In the materials and methods section, the significance chosen by the authors should be indicated (p<0.05 or another). Various significances are indicated in the tables, up to p<0.10. Which one do the authors choose?

Author Response

Point 1: Allow me to make a series of comments to the authors, especially referring to the materials and methods section. This section should describe how the sample size was established and the sample was selected, but the results of the sample should be included in the results section. Thus, the data that appears between the lines of the text 101 to 108 and Table 1 must be detailed in the results section.

Response 1: This section has been moved from the materials and methods to the results section as advised.

Point 2: In the materials and methods section, the significance chosen by the authors should be indicated (p<0.05 or another). Various significances are indicated in the tables, up to p<0.10. Which one do the authors choose?

Response 2: Thank you for the comment. Given the exploratory nature of the study, we decided to have a more lenient approach to bivariate associates of PTSS/PTSD-AA for entry into the multivariate model. However, as this didn’t really change the variables that we included, we have removed the reference to p<.10 for clarity.

Reviewer 2 Report

I enjoyed reading this paper on predictors of PTSS/PTSD at 6 months, controlling for T1 PTSS (2-4 weeks after PICU admission). Major findings included the effects of ketamine, and cognitive and affective experiences during PICU. A mediation model suggests that cognitive processes may mitigate affective effects on PTSS at 6 months.    

1. line 122: pls clarify meaning of other patient death variable. 

2. Please add sample sizes to titles of Tables 1-3.

3. Please confirm the table headers for Table 2. Also clarify meaning of "Exposed to distressing event in 

PICU" variable.

4. Table 4: add a note that T1 PTSS = acute PTSS

5. Table 6 and Fig. 1: please report standardised coefficients

6. lines 287-90: please comment on indications for ketamine in a PICU setting and how it may be administered. Is it used in more severe cases? If so, this may be another interpretation of its effect on PTSS.

7. lines 330-36: perhaps an implication may be that the use of ketamine may also increase the need to provide psychological support to promote adaptive cognitive-affective processing. Regardless, are there any clinical approaches that the authors may recommend to support such processing in children?  

8. Clarify meaning of "PIM2 Risk of Mortality" variable.

9. I may have missed this but I had trouble finding descriptive statistics on acute PTSS, PTSS at 6 months, and the PTSD-AA at 6 months diagnosis percentage.

Author Response

Point 1: line 122: pls clarify meaning of other patient death variable. 

Response 1: We accessed hospital records to determine whether another patient died in the unit during the child’s admission, in addition to asking parents a more general question about whether their child was exposed to a distressing event while they were in PICU. We wanted to specifically look at other patient death as a potentially traumatic event that is more likely to occur in the PICU compared to general wards. This has been clarified in the text in lines 108-109.

Point 2: Please add sample sizes to titles of Tables 1-3.

Response 2: Added

Point 3: Please confirm the table headers for Table 2. Also clarify meaning of "Exposed to distressing event in PICU" variable.

Response 3: We have updated table headings for Table 2. We have also now put all descriptive statistics into Table 1, and reported just the bivariate correlations in Table 2 for clarity. We have also clarified the “exposed to distressing event in PICU” to “parent’s report of child exposure to any distressing events in PICU” in line 107.

Point 4: Table 4: add a note that T1 PTSS = acute PTSS

Response 4: Thank you for drawing attention to this. We have renamed T1 PTSS to Acute PTSS throughout for consistency.

Point 5: Table 6 and Fig. 1: please report standardised coefficients

Response 5: Thank you for this thoughtful comment. This mediation analysis was conducted using Hayes PROCESS Macro for SPSS prior to 2021, which only produced unstandardized regression coefficients. Although newer versions of the PROCESS macro will provide standardised coefficients, we note that Hayes still recommends the use of unstandardised over standardised coefficients (Hayes, 2017). For example, Hayes points out that standardized coefficients do not allow a comparison of group means and relative indirect effects straightforwardly, and if the key variables in an empirical model are based on similar measurement scales, then reporting unstandardized coefficients gives a straightforward interpretation to all coefficients and effect sizes and there is no advantage in reporting standardized coefficients.

Point 6: lines 287-90: please comment on indications for ketamine in a PICU setting and how it may be administered. Is it used in more severe cases? If so, this may be another interpretation of its effect on PTSS.

Response 6: We have included some more information on the administration and indications of ketamine use in PICU. Use of ketamine isn’t necessarily indicative of severity of illness or particular diagnostic groups, and given that PIM2 risk of death, PICU length of stay and mechanical ventilation were not related to PTSS, we hypothesize that the emergence reactions might explain ketamine’s influence on PTSS.

Point 7:. lines 330-36: perhaps an implication may be that the use of ketamine may also increase the need to provide psychological support to promote adaptive cognitive-affective processing. Regardless, are there any clinical approaches that the authors may recommend to support such processing in children?  

Response 7: We have added some specific approaches that may help children process their admission on lines 345 -349.

Point 8. Clarify meaning of "PIM2 Risk of Mortality" variable.

Response 8: Further information about PIM2 has been added on lines 115-117.

Point 9: I may have missed this but I had trouble finding descriptive statistics on acute PTSS, PTSS at 6 months, and the PTSD-AA at 6 months diagnosis percentage.

Response 9: Thank you for this comment. We have added text in the results regarding descriptive stats for PTSS at 6 months and PTSD-AA, and we have added Acute PTSS to Table 1.

Round 2

Reviewer 1 Report

I thank the authors for their response to the reviewer's suggestions.

Reviewer 2 Report

No further comments.